Identification of differentially expressed key genes between glioblastoma and low-grade glioma by bioinformatics analysis

Xu Yang 1 2
http://orcid.org/0000-0002-9442-7202 Geng Rongxin 1 2
Yuan Fan’en 1 2
http://orcid.org/0000-0003-3659-3242 Sun Qian 1 2
http://orcid.org/0000-0002-9413-1030 Liu Baohui 1 2 bliu666@whu.edu.cn
http://orcid.org/0000-0002-9413-1030 Chen Qianxue 1 2 chenqx666@whu.edu.cn
1 Department of Neurosurgery, Renmin Hospital of Wuhan University , Wuhan, Hubei , China
2 Brain Tumor Clinical Center of Wuhan , Wuhan, Hubei , China
Abdullah Jafri
Electronic publication date: 2019 Mar 7
Publication date: 2019
Volume: 7
Electronic Location ID: e6560
Received 2018 Oct 24; Accepted 2019 Feb 4
Copyright: © 2019 Xu et al.
Copyright year: 2019
Copyright holder: Xu et al.
License: This is an open access article distributed under the terms of the Creative Commons Attribution License, which permits unrestricted use, distribution, reproduction and adaptation in any medium and for any purpose provided that it is properly attributed. For attribution, the original author(s), title, publication source (PeerJ) and either DOI or URL of the article must be cited.
License URL: https://creativecommons.org/licenses/by/4.0/

Keywords: Key genes, Glioblastoma, Low grade glioma, Bioinformatics analysis, Biomarkers, SAA1, TIMP1

Funding: National Natural Science Foundation of China 81572489, 81372683, and 81502175 This work was supported by the National Natural Science Foundation of China (No. 81572489, 81372683, and 81502175). The funders had no role in study design, data collection and analysis, decision to publish, or preparation of the manuscript.

==============================
Gliomas are a very diverse group of brain tumors that are most commonly primary tumor and difficult to cure in central nervous system. It’s necessary to distinguish low-grade tumors from high-grade tumors by understanding the molecular basis of different grades of glioma, which is an important step in defining new biomarkers and therapeutic strategies. We have chosen the gene expression profile GSE52009 from gene expression omnibus (GEO) database to detect important differential genes. GSE52009 contains 120 samples, including 60 WHO II samples and 24 WHO IV samples that were selected in our analysis. We used the GEO2R tool to pick out differently expressed genes (DEGs) between low-grade glioma and high-grade glioma, and then we used the database for annotation, visualization and integrated discovery to perform gene ontology analysis and Kyoto encyclopedia of gene and genome pathway analysis. Furthermore, we used the Cytoscape search tool for the retrieval of interacting genes with molecular complex detection plug-in applied to achieve the visualization of protein–protein interaction (PPI). We selected 15 hub genes with higher degrees of connectivity, including tissue inhibitors metalloproteinases-1 and serum amyloid A1; additionally, we used GSE53733 containing 70 glioblastoma samples to conduct Gene Set Enrichment Analysis. In conclusion, our bioinformatics analysis showed that DEGs and hub genes may be defined as new biomarkers for diagnosis and for guiding the therapeutic strategies of glioblastoma.

Introduction

Gliomas are a very diverse group of brain tumors that are most commonly primary tumor and difficult to cure in central nervous system (Louis et al., 2016; Ostrom et al., 2017). They are classified according to their clinical and histopathological characteristics in four grades, including low-grade gliomas–(1) grade I astrocytomas—pilocytic astrocytomas, (2) grade II diffuse astrocytomas, and (3) grade II oligodendrogliomas; High-grade gliomas–(1) Grade III anaplastic astrocytomas, (2) Grade III anaplastic oligodendrogliomas, and (3) grade IV glioblastomas multiforme (GBM) (Sriram & Huse, 2015). Low-grade gliomas (LGG) (astrocytomas, oligodendrogliomas, and oligoastrocytomas) are considered relatively benign, well-differentiated tumors and have 5-year survival rates of 59.9% (Claus & Black, 2006). GBM is the most common primary malignant brain tumor in adults (Ramos, Magge & Ramakrishna, 2018). Despite multiple therapeutic strategies, including surgery, radiation, and chemotherapy, the average survival time of GBM patients is less than 15 months (Liu et al., 2016). Additionally, approximately 70% of LGG patients develop GBM within 5–10 years (Furnari et al., 2007). With the development of molecular pathology in gliomas, several biomarkers are routinely applied to evaluate gliomas including O-6-methylguanine-DNA methyltransferase (MGMT) promoter methylation, EGFR alterations, Isocitrate dehydrogenase 1 (IDH1) or Isocitrate dehydrogenase 2 (IDH2) mutations, and 1p19q co-deletion as many of these markers have become standard of care for molecular testing and prerequisites for clinical trial enrollment (Rodriguez, Vizcaino & Lin, 2016). Therefore, it’s necessary to differentiate LGG to GBM by understanding the molecular basis of different grades of glioma, which is an important step in defining new biomarkers and therapeutic strategies.

Gene expression profiling analysis is a useful method with broad clinical application for identifying tumor-related genes in various types of cancer, from molecular diagnosis to pathological classification, from therapeutic evaluation to prognosis prediction, and from drug sensitivity to neoplasm recurrence (De Preter et al., 2010; Freije et al., 2004; Kim et al., 2011; Kulasingam & Diamandis, 2008). In recent years, large scales of gene profiling have been made to identify the overwhelming number of genes by the use of microarrays in clinical practice, and complicated and systemic statistical analyses should be made to allow both repeatability and independent validation (Cheng et al., 2016).

In this analysis, GEO2R online tool was applied to look for the differentially expressed genes (DEGs) according to GSE52009 from GEO. Afterwards, we produced a heatmap and picked out 15 genes with higher degree of connectivity from the DEGs selected. Thereafter, we analyzed the cellular component (CC), biological process (BP), molecular function (MF), and Kyoto Encyclopedia of Gene and Genome (KEGG) pathways of the DEGs. In addition, the overall survival (OS) analysis and expression of these hub genes were made online. Then, we established protein–protein interaction (PPI) network of the DEGs and managed a gene set enrichment analysis (GSEA) using GBM patient gene profiling data (GSE53733).

Materials and Methods

Data of microarray

Gene expression profile of GSE52009, GSE53733, and GSE4290 were downloaded from GEO database, which is a public and freely accessible database. Based on Agilent GPL6480 platform (Agilent-014850 Whole Human Genome Microarray 4x44K G4112F), GSE52009 dataset included 120 samples, containing 60 WHO II samples and 24 WHO IV samples. GSE53733 was based on the GPL570 platform ((HG-U133_Plus_2) Affymetrix Human Genome U133 Plus 2.0 Array), which contained 70 GBM samples. GSE4290 was based on the GPL570 platform ((HG-U133_Plus_2) Affymetrix Human Genome U133 Plus 2.0 Array), which included 180 samples, containing 76 WHO II samples and 81 WHO IV samples.

Screen genes of differential expression

Differentially expressed genes between low-grade glioma and high-grade glioma was detected by GEO2R, which was an online analysis tool based on R language (Davis & Meltzer, 2007). We set the adjust P-value < 0.05 and |logFC| ≥ 2 as the selection criteria to decrease the false positive rate and false discovery rate. Furthermore, the top 15 genes with higher degree of connectivity were selected as hub genes among the 133 discovered DEGs which includes 56 downregulated genes and 77 upregulated genes. In addition, we used visual hierarchical cluster analysis to show the two groups by Morpheus online analysis software (https://software.broadinstitute.org/morpheus/) and volcano plot of two groups by ImageGP (http://www.ehbio.com/ImageGP/index.php/Home/Index/index.html) after the relative raw data of TXT files were downloaded.

Gene ontology and KEGG pathway analysis of DEGs

With functions including MF, biological pathways, and CC, gene ontology (GO) analysis we annotated genes and gene products (Gene Ontology Consortium, 2006). KEGG comprises a set of genome and enzymatic approaches and abiological chemical energy online database (Kanehisa & Goto, 2000). It is a resource for systematic analysis of gene function and related high-level genome functional information. Database for annotation, visualization and integrated discovery (DAVID) (https://david.ncifcrf.gov/) can provide systematic and comprehensive biological function annotation information for high-throughput gene expression (Dennis et al., 2003). Therefore, we applied GO and KEGG pathway analyses to the DEGs by using DAVID online tools at functional level. We considered P < 0.05 had significant differences. In addition, we used visual analysis to show GO Enrichment plot of two groups by ImageGP (http://www.ehbio.com/ImageGP/index.php/Home/Index/index.html) after the relative raw data of TXT files were downloaded (Geng et al., 2018).

PPI network and module analysis

The online tool, search tool for the retrieval of interacting genes (STRING), is designed to demonstrate the interaction between different proteins (Szklarczyk et al., 2015). STRING in Cytoscape was applied and mapped the DEGs into STRING to detect the possible relationship among the selected DEGs. We set the confidence score ≥0.4, maximum number of interactors = 0 as the selection criteria. In addition, the molecular complex detection (MCODE) was used to screen modules of PPI network in Cytoscape with degree cutoff = 2, node score cutoff = 0.2, k-core = 2, and max. depth = 100. DAVID was used to perform the signal pathway analysis of genes in the module. A total of 15 hub genes were also mapped into STRING with confidence score ≥0.4, maximum number of interactors ≤5. The potential information was explored through GO and KEGG pathway analysis.

Comparing the expression level of the hub genes

GlioVis (http://gliovis.bioinfo.cnio.es/) is a user-friendly web application for data visualization and analysis to explore brain tumors expression datasets, which was used to analyze the gene expression data of brain tumors and normal samples based on the TCGA datasets. (Bowman et al., 2017). The customizable functions are provided such as analyzing the differences of expression levels between glioblastoma and low-grade glioma, so the expression of these genes was demonstrated. And the relationship could be visualized through the boxplot. All values are presented as the mean ± SD. All statistical analyses were performed by SPSS 19.0 software. A difference of P < 0.05 was considered statistically significant.

Gene expression profile and gene set enrichment analysis

The expression profiles of GSE53733 were downloaded from the GEO data base. We used GSEA (http://www.broadinstitute.org/gsea) to detect the potential genes influenced by Serum amyloid A1 (SAA1) and tissue inhibitor of metalloproteinases-1 (TIMP1) through Java programming. According to their hub genes expression level (top 50%: high vs. bottom 50%: low), we divided the patients into two groups, and GSEA was conducted to analyze the effects of selected genes expression level on different BP. We set P-value of <0.05 and false discovery rates of <0.25 as selection criteria to confirm significant gene sets.

Human tissue samples

Low-grade gliomas and GBM tissues were collected from the Department of Neurosurgery, Renmin Hospital of Wuhan University, Wuhan, China. The clinical glioma specimens were examined and diagnosed by pathologists at Renmin Hospital of Wuhan University. This study was approved by the Institutional Ethics Committee of the Faculty of Medicine at Renmin Hospital of Wuhan University (approval number: 2012LKSZ (010) H). Informed consent was obtained from all patients whose tissues were used.

RNA extraction and quantitative real-time PCR

Total RNA from cancer tissues was prepared using Trizol reagent (Invitrogen, Carlsbad, CA, USA), and cDNA was synthesized using a PrimeScript RT Reagent Kit with gDNA Eraser (RR047A; Takara, Kusatsu, Japan). Quantitative real-time PCR (qPCR) for SAA1 and TIMP1 mRNA levels were performed using SYBR Premix Ex Taq II (RR820A, Takara, Kusatsu, Japan) according to the manufacturer’s instructions and performed in Bio-Rad CFX Manager 2.1 real-time PCR Systems (Bio-Rad, Hercules, CA, USA). GAPDH was used as internal controls. The data were analyzed by the relative Ct method and expressed as a fold change compared with the control. The primer sequences included the following: GAPDH 5′-GGAGCGAGATCCCTCCAAAAT-3′(Forward), 5′-GGCTGTTGTCATACTTCTCATGG-3′(Reverse); SAA1 5′-CCTGGGCTGCAGAAGTGATCAGCGA-3′(Forward), 5′-AGTCCTCCGCACCATGGCCAAAGAA-3′(Reverse); TIMP1 5′-CTTCTGCAATTCCGACCTCGT-3′(Forward), 5′-ACGCTGGTATAAGGTGGTCTG-3′(Reverse).

Results

Identification of DEGs and hub genes

A total of 60 WHO II samples and 24 WHO IV samples from GSE52009 were selected in this study. DEGs were detected by applying the GEO2R online analysis tool, setting adjust P-value < 0.05 and |logFC| ≥ 2 as selection criteria. A total of 133 differential expressed genes, containing 77 upregulated genes and 56 downregulated genes, were detected after the analysis of GSE52009. In addition, we selected 15 hub genes with higher degree of connectivity (Table 1). The results were validated with a DEG expression heatmap and volcano plot of the all downregulated genes and upregulated genes (Fig. 1).

Table 1 Top 15 hub genes with higher degree of connectivity.

Gene	Degree	P-value	
VEGFA	13	7.55E-07	
NDC80	8	7.16E-09	
IL8	8	5.15E-07	
CENPA	7	3.31E-09	
CENPF	7	6.06E-10	
NCAPG	7	9.09E-10	
ASPM	7	1.86E-08	
RRM2	7	1.13E-09	
ITGA2	6	2.45E-09	
ANXA1	6	6.32E-08	
CDCA2	6	6.42E-09	
PLAT	5	2.64E-08	
PARPBP	5	6.34E-15	
TIMP1	4	2.06E-06	
SAA1	4	2.40E-06	

Figure 1 Differentially expressed gene expression heatmap and volcano plot of glioma.

(A) Differentially expressed gene expression heatmap of glioma (all upregulated and downregulated genes). (B) Differentially expressed genes were selected by volcano plot filtering (fold change ≥ 1 and P-value ≤ 0.05). The blue point in the plot represents the differentially expressed genes with statistical significance.

GO function and KEGG pathway enrichment analysis

To explore more particular knowledge of the selected DEGs, we used DAVID to gain the results of GO function and KEGG pathway enrichment analysis. All DEGs were imported to DAVID software, and GO analysis results demonstrated that upregulated and downregulated DEGs were particularly enriched in the following biological processes (BP): cell migration, locomotion and leukocyte migration, cell motility for upregulated DEGs, and for downregulated DEGs nervous system development, brain development and regulation of cell projection organization (Table 2; Figs. 2A and 2B). The upregulated DEGs were enriched in phospholipase A2 inhibitor activity, growth factor binding, extracellular matrix (ECM) structural construction, receptor binding, and the downregulated DEGs were enriched in calcium ion binding, structural construction of myelin sheath, and protein complex binding for MF (Table 2; Fig. 2). Moreover, GO CC analysis showed that the upregulated DEGs were enriched in the proteinaceous ECM, ECM and cytoplasmic membrane-bounded vesicle lumen, and downregulated DEGs enriched in neuron part, myelin sheath, and internode region of axon (Table 2; Figs. 2A and 2B).

Table 2 Gene ontology analysis of differentially expressed genes associated with LGG and HGG.

Expression	Category	Term	Count	%	P-value	FDR	
Upregulated	GOTERM_BP_FAT	GO:0016477—cell migration	13	22.80702	3.95E-05	0.067618	
GOTERM_BP_FAT	GO:0050900—leukocyte migration	8	14.03509	5.28E-05	0.090452	
GOTERM_BP_FAT	GO:0040011—locomotion	14	24.5614	1.09E-04	0.185762	
GOTERM_BP_FAT	GO:0051674—localization of cell	13	22.80702	1.22E-04	0.208673	
GOTERM_BP_FAT	GO:0048870—cell motility	13	22.80702	1.22E-04	0.208673	
GOTERM_MF_FAT	GO:0019838—growth factor binding	4	7.017544	0.004366	5.428929	
GOTERM_MF_FAT	GO:0019834—phospholipase A2 inhibitor activity	2	3.508772	0.010298	12.37092	
GOTERM_MF_FAT	GO:0005102—receptor binding	10	17.54386	0.010815	12.95228	
GOTERM_MF_FAT	GO:0005201—extracellular matrix structural constituent	3	5.263158	0.018986	21.69308	
GOTERM_MF_FAT	GO:0005125—cytokine activity	4	7.017544	0.019477	22.19193	
GOTERM_CC_FAT	GO:0005615—extracellular space	16	28.07018	3.09E-06	0.003877	
GOTERM_CC_FAT	GO:0005578—proteinaceous extracellular matrix	7	12.2807	4.24E-04	0.531176	
GOTERM_CC_FAT	GO:0031012—extracellular matrix	8	14.03509	5.41E-04	0.676623	
GOTERM_CC_FAT	GO:0060205—cytoplasmic membrane-bounded vesicle lumen	4	7.017544	0.002985	3.680925	
GOTERM_CC_FAT	GO:0031983—vesicle lumen	4	7.017544	0.003066	3.779192	
Downregulated	GOTERM_BP_FAT	GO:0007399—nervous system development	15	31.91489	3.29E-04	0.533895	
GOTERM_BP_FAT	GO:0007420—brain development	7	14.89362	0.00548	8.553316	
GOTERM_BP_FAT	GO:0060322—head development	7	14.89362	0.007003	10.80691	
GOTERM_BP_FAT	GO:0031344—regulation of cell projection organization	6	12.76596	0.011466	17.11031	
GOTERM_BP_FAT	GO:0007417—central nervous system development	7	14.89362	0.019918	27.91963	
GOTERM_MF_FAT	GO:0005509—calcium ion binding	6	12.76596	0.021579	24.70502	
GOTERM_MF_FAT	GO:0019911—structural constituent of myelin sheath	2	4.255319	0.02239	25.5132	
GOTERM_MF_FAT	GO:0032403—protein complex binding	5	10.6383	0.092825	71.83703	
GOTERM_CC_FAT	GO:0097458—neuron part	15	31.91489	6.61E-06	0.007993	
GOTERM_CC_FAT	GO:0045202—synapse	9	19.14894	8.96E-04	1.078808	
GOTERM_CC_FAT	GO:0043005—neuron projection	9	19.14894	0.00424	5.011629	
GOTERM_CC_FAT	GO:0033269—internode region of axon	2	4.255319	0.01097	12.49482	
GOTERM_CC_FAT	GO:0043209—myelin sheath	4	8.510638	0.01206	13.65505	

Figure 2 GO analysis results of DEGs.

(A) and downregulated DEGs (B) were particularly enriched in BP, MF, and CC. (C) The most significantly enriched KEGG pathway of the upregulated DEGs. GO, gene ontology; BP, biological process; MF, molecular function; CC, cell component; KEGG, Kyoto Encyclopedia of Genes and Genomes.

Interestingly, the most significantly enriched of KEGG pathway only showed in upregulated pathway, including adhesion, ECM-receptor interaction, amoebiasis, and PI3K-Akt signaling pathway (Table 3; Fig. 2C).

Table 3 KEGG pathway analysis of differentially expressed genes associated with HGG.

Expression	Category	Term	Count	%	P-value	Genes	FDR	
Upregulated	KEGG_PATHWAY	hsa04510:Focal adhesion	4	7.017544	0.007454	LAMB1, VEGFA, ITGA2, COL1A1	7.094772	
KEGG_PATHWAY	hsa04512:ECM-receptor interaction	3	5.263158	0.012926	LAMB1, ITGA2, COL1A1	12.01164	
KEGG_PATHWAY	hsa05146:Amoebiasis	3	5.263158	0.018811	LAMB1, CXCL8, COL1A1	17.03775	
KEGG_PATHWAY	hsa04151:PI3K-Akt signaling pathway	4	7.017544	0.02979	LAMB1, VEGFA, ITGA2, COL1A1	25.72978	
KEGG_PATHWAY	hsa05200:Pathways in cancer	4	7.017544	0.041599	LAMB1, VEGFA, ITGA2, CXCL8	34.15806	

Hub genes and module screening from PPI network

Protein–protein interaction network of the top 15 hub genes with higher degree of connectivity was made based on the information in the STRING protein query from public databases (Fig. 3A). The top module was selected by using MCODE plug-in in the PPI network (Fig. 3B).

Figure 3 protein–protein interaction network and top module of hub genes.

(A) The protein–protein interaction network of the top 15 hub genes. (B) Top module from the protein–protein interaction network.

The Kaplan–Meier plotter of hub genes

The website, http://gepia.cancer-pku.cn/, freely provides the prognostic data of the hub genes. It was found that expression of VEGFA (HR 4.2, P < 0.001) was associated with worse OS for glioblastoma patients, as well as NDC80 (HR 5.8, P < 0.001), CENPA (HR 5.3, P < 0.001), CENPF (HR 3.9, P < 0.001), Non-SMC condensin I complex subunit G (NCAPG) (HR 5.6, P < 0.001), ASPM (HR 5, P < 0.001), ITGA2 (HR 3, P < 0.001), TIMP1 (HR 7, P < 0.001)and SAA1 (HR 4.8, P < 0.001) (Fig. 4).

Figure 4 Prognostic value of hub genes in glioma patients.

Prognostic value of hub genes (VEGFA, NDC80, CENPA, CENPF, NCAPG, ASPM, ITGA2, TIMP1, and SAA1) in glioma patients. HR, hazard ratio. (A) VEGFA (HR 4.2, P < 0.001) was associated with worse OS for glioblastoma patients; (B) NDC80 (HR 5.8, P < 0.001) was associated with worse OS for glioblastoma patients; (C) CENPA (HR 5.3, P < 0.001) was associated with worse OS for glioblastoma patients; (D) CENPF (HR 3.9, P < 0.001) was associated with worse OS for glioblastoma patients; (E) NCAPG (HR 5.6, P < 0.001) was associated with worse OS for glioblastoma patients; (F) ASPM (HR 5, P < 0.001) was associated with worse OS for glioblastoma patients; (G) ITGA2 (HR 3, P < 0.001) was associated with worse OS for glioblastoma patients; (H) TIMP1 (HR 7, P < 0.001) was associated with worse OS for glioblastoma patients; (I) SAA1 (HR 4.8, P < 0.001) was associated with worse OS for glioblastoma patients.

Expression level and relationship with molecular pathologic diagnosis of hub genes

We used data from GlioVis to detect the hub gene expression level between GBM and LGG including astrocytoma, oligodendroglioma, and oligoastrocytoma, the expression level of SAA1 and TIMP1 significantly increased in GBM (Figs. 5A and 5C). The expression levels of SAA1 have no significant difference in three kind of LGG (Fig. 5B). However, the expression level of TIMP1 is significantly higher in astrocytoma than oligodendroglioma and oligoastrocytoma (Fig. 5D). We further verified our finding in the GSE4290 dataset and got consistent result (Fig. S1). Then we detected the sample collected in our hospital and found both SAA1 and TIMP1 are significantly increased in GBMs compared with LGG (Fig. S2; Table S1). We also detect the relationship between expression level and molecular pathologic diagnosis of hub genes. We found both SAA1 and TIMP1 increase in both Isocitrate dehydrogenase (IDH) mutant IDH wild type. The same results were found in MGMT promoter and non-deletion of chromosome 1p.19q. Because of the limited samples in the datasets, we didn’t the result of co-deletion of chromosome 1p.19q. (Figs. 5E and 5F) Further, we also found that both SAA1 and TIMP1 played important roles in MES-like in the Isocitrate dehydrogenase (IDH) wild type (Figs. 5G and 5H).

Figure 5 The expression level and potential function of SAA1 and TIMP1.

(A) SAA1 significantly increased in glioblastomas; (B) The expression level of SAA1 have no significant difference in LGG; (C) TIMP1 significantly increased in glioblastomas; (D) TIMP1 is significantly higher in astrocytoma than oligodendroglioma and oligoastrocytoma; (E) and (F) SAA1 and TIMP1 increase in both IDH mutant and IDH wild type. The same results were found in MGMT promoter and non-deletion of chromosome 1p.19q; (G) and (H) SAA1 and TIMP1 played important roles in MES-like in the IDH wild type; (I) and (J) SAA1 regulates biology process associated with inflammatory response processes and cytokine mediated signaling pathway; (K) and (L) TIMP1 negatively regulates adaptive immune response based on somatic recombination of immune receptors built from a leucine-rich superfamily and TIMP1 also negatively regulates response to interferons.

Gene expression profile and gene set enrichment analysis

We managed a GSEA by using GBM patient gene profiling data (GSE53733), and showed in Fig. 5, gene set differences in SAA1 in low versus high glioma patients indicated that SAA1 regulates biology process mainly associated with inflammatory response processes (P < 0.001 FDR = 0.012) and cytokine mediated signaling pathway (P < 0.001 FDR = 0.012) (Figs. 5I and 5J). We considered that SAA1 may negatively regulate inflammatory response and might promote the survival of cancer cells. We concluded that TIMP1 might negatively regulates adaptive immune response based on somatic recombination of immune receptors built from a leucine-rich superfamily (P < 0.001 FDR = 0.021) and response to interferon (P < 0.001 FDR = 0.027) may promote the survival of cancer cells (Figs. 5K and 5L).

Discussion

In this study, we identified 15 DEGs between GBM and LGG and used a series of bioinformatics analyses to screen the key genes and pathways associated with glioma. GSE52009 dataset contains 60 WHO II samples and 24 WHO IV samples. In order to improve the statistical power of DEGs, we defined that the absolute value of the logarithm (base 2) fold change (logFC) greater than 2 and 133 DEGs were obtained. Bioinformatics analysis on DEGs, including GO enrichment, KEGG pathway, PPI network, and survival analysis, expression level, gene set enrichment analysis, found GBM genes and pathways that play important roles in glioma development.

The GO analysis showed that the upregulated DEGs were mainly associated with cell migration, ECM structural construction, cell motility and downregulated DEGs were involved central nervous system development, calcium ion binding and internode region of axon. Additionally, the KEGG pathways of upregulated DEGs regulate focal adhesion, ECM-receptor interaction, PI3K-Akt signaling pathway. Among these DEGs, we selected 15 hub genes with higher degree of connectivity. In addition, we found several hub genes with worse OS and higher expression level in glioma patients, including VEGFA, NDC80, TIMP1, SAA1, CENPA, CENPF, and NCAPG and we firstly found relationship of SAA1, TIMP1, and molecular pathology in GBM. We could hypothesize that these genes might contribute to the malignance of glioma and SAA1 and TIMP-1 may be biomarkers in GBM.

Glioblastomas multiformes are highly vascularized cancers with high levels of VEGF and VEGF-A seems to be the most critical one, mainly operating in the activation of quiescent endothelial cells and promoting cell migration and proliferation (Plate, Scholz & Dumont, 2012). NDC80 is a mitotic regulator and a major element of outer kinetochore which has been reported to drive functions in assembly checkpoint and chromosome segregation of mitosis regulation. NDC80 was mainly enriched in proliferation and procession of cancer in previous studies (Suzuki, Badger & Salmon, 2015). Addition, a recently study demonstrated that the expression of NDC80 in HEB was significantly lower than in GBM cell lines and had a negative correlation with the prognosis of patients (Zhong et al., 2018). Interleukin (IL)-8 is a chemokine which was upregulated by NF-κB in GBMs and promotes a more aggressive phenotype mostly through the enhancement of angiogenesis and cell migration. More and more evidence demonstrated that the IL-8 molecular pathways will allow the generation of both novel therapeutic approaches and diagnostic tools (Kosmopoulos et al., 2017). NCAPG8 is a novel mitotic gene for cell proliferation and migration, which has been less well studied in cancers, and a recently study demonstrated that NCAPG over expressed in GBMs and promote cell proliferation (Liang et al., 2016).

Serum amyloid A1 which was secreted by liver is an acute-phase high density lipo-protein in immune response. Injury, inflammation, and brain trauma can elevate the plasma levels of SAA1 (Lu et al., 2014; Villapol et al., 2015). Further, it has long been suspected that SAA1 might be a prognostic marker and predictor of cancer risk. Elevated levels of SAA1 in the serum of cancer patients directly correlate with poor prognosis and tumor aggressiveness in various types of cancer, including lung cancer (Cho et al., 2010), cell renal carcinoma (Kosari et al., 2005), melanomas (Findeisen et al., 2009) and so on. Normal brain does not express SAA1 (Liang et al., 1997), though an in vitro study demonstrated that microglia and astrocytes are responsive to SAA (Yu et al., 2014). Recently, it has been reported that SAA1 the expression levels in GBM patients are upregulated on both mRNA and protein in human GBMs, and SAA1 involves in angiogenesis via HIF-1α and tumor associated macrophages. Serum levels of SAA1 were associated with the grades of gliomas but did not affect the clinical outcomes of patients with GBM (Knebel et al., 2017). Consistently, SAA1 has been reported as a molecular/metabolic signature that can help identify patients are at high risk of malignant GBM and promotes glioma cells migration and invasion through integrin aVb3 (Lin et al., 2018). Further, although it’s unknown why SAA1 upregulated in GBM and other malignant cancers, it has been speculated that SAA proteins play a primary role in the regulation of immunity and invasion processes (Moshkovskii, 2012), which is consistent with the result of our study. Thus, we hypothesis SAA1 could be a biomarker of GBM and predict the prognosis of GBM patients. The mechanism of SAA1 regulate in GBM need further research.

The tissue inhibitors of metalloproteinases (TIMPs, including TIMP-1, TIMP-2, TIMP-3, TIMP-4) are well known play critical roles in both metastasis and invasion through ECM remodeling which are controlled by the activity of matrix metalloproteinases (MMPs) (Jackson et al., 2017; Ries, 2014). Among the four well-known TIMPs characterized so far, most focus has been on TIMP-1 defined as a naturally occurring inhibitor of most MMPs, a family of zinc dependent endopeptidases essential for degrading components of the ECM (Aaberg-Jessen et al., 2009). In addition, TIMP1 shows protease-independent function including anti-apoptotic, antiangiogenic, and differentiation activities in cells (Bridoux et al., 2013; Mandel et al., 2017). Over the past year, more and more studies have focused on the influence of TIMP1 in cancers. Serum or urine levels of TIMP1 are also considered as a diagnostic predictor in pancreatic ductal carcinomas containing extensive desmoplasia (Jenkinson et al., 2015; Roy et al., 2014). Increased levels of cytosolic TIMP1 in pretreatment tumor tissue is associated with a significantly shorter OS in patients with breast cancer receiving standard adjuvant chemotherapy (Dechaphunkul et al., 2012). It has also been reported that low expression of TIMP-1 in glioblastoma patient predicts longer survival. The shorter survival of glioblastoma patients with a high tumor TIMP-1 level may be explained by the antiapoptotic effect of TIMP-1 preventing apoptosis induced by radiation and chemotherapy (Aaberg-Jessen et al., 2009). More recently, Aaberg-Jessen et al. (2018) demonstrated that Co-expression of TIMP-1 and CD63 might have effects in glioblastoma stemness and may predict the poor prognosis of patients through influencing tumor aggressiveness and resistance of therapy. We consider that TIMP-1 can be identified as a future biomarker for prognosis or monitoring patients’ treatment response. However, all these studies didn’t demonstrate the specific mechanism, which is the direction for further researches.

Additionally, large-scale efforts aimed at characterizing the genomic alterations in human glioblastoma, however, these efforts helped to clarify the role of genomic alterations in the pathogenesis of glioblastoma but were not designed to address intratumor heterogeneity. Recently, Puchalski et al. (2018) described the Ivy Glioblastoma Atlas in which we have assigned key genomic alterations and gene expression profiles to the tumor’s anatomic features. The anatomic feature included the leading edge (LE), infiltrating tumor (IT), cellular tumor (CT). We also found that the expression levels of SAA1 and TIPM-1 were higher in IT and CT than LE from the atlas. However, the specific mechanisms of these differences aren't presently clear; therefore, we need to do further research in the future.

Conclusion

We found that these key genes (identified by a series of bioinformatics analyses on DEGs between glioblastoma samples and low-grade glioma samples) were most likely related to the development of glioma. These hub genes could also affect the survival time of glioma patients. These identified genes and pathways provide a more detailed molecular mechanism for underlying glioma initiation and development. According to the study, SAA1 and TIMP1 can be considered as biomarkers or therapeutic targets for monitoring patient treatment response for glioblastoma. However, further molecular and biological experiments are required to confirm the functions of the key genes in glioblastoma.

Supplemental Information

Supplemental Information 1 The expression of SAA1 and TIMP1 in GSE4290 Dataset.

Click here for additional data file.

Supplemental Information 2 The expression of SAA1 and TIMP1 in Human tissues.

Figure S2: (A) SAA1 significantly increased in glioblastomas; (B) TIMP1 significantly increased in glioblastomas.

Click here for additional data file.

Supplemental Information 3 Demographicdata from patients analyzed in this study.

LGG: low grade glioma; GBM: glioblastoma; M: male; F: female; *Age at diagnosis was calculated from date of birth to date of surgery.

Click here for additional data file.

Additional Information and Declarations

Competing Interests

Author Contributions

Human Ethics

Data Availability

The authors declare that they have no competing interests.

Yang Xu conceived and designed the experiments, analyzed the data, contributed reagents/materials/analysis tools, prepared figures and/or tables.

Rongxin Geng analyzed the data, contributed reagents/materials/analysis tools, prepared figures and/or tables.

Fan’en Yuan analyzed the data, contributed reagents/materials/analysis tools, prepared figures and/or tables.

Qian Sun analyzed the data, contributed reagents/materials/analysis tools.

Baohui Liu conceived and designed the experiments, contributed reagents/materials/analysis tools, authored or reviewed drafts of the paper, approved the final draft.

Qianxue Chen conceived and designed the experiments, contributed reagents/materials/analysis tools, authored or reviewed drafts of the paper, approved the final draft.

The following information was supplied relating to ethical approvals (i.e., approving body and any reference numbers):

Institutional Ethics Committee of the Faculty of Medicine at Renmin Hospital of Wuhan University approval (2012LKSZ (010) H) to carry out the study within its facilities.

The following information was supplied regarding data availability:

Data is available at NCBI GEO, accession numbers: GSE52009, GSE4290, GSE53733.

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
