# Peer review of "Identification of differentially expressed key genes between glioblastoma and low-grade glioma by bioinformatics analysis"

_PeerJ, doi:10.7717/peerj.6560_

## Round 0.1 · original submission · Major Revisions

Dear Authors,

You will need to do a major revisions to your manuscript as per the reviewers suggestions. The experimental design and the statistics are extremely important.

Reviewer 1 ·

Basic reporting

No comment.

Experimental design

No comment.

Validity of the findings

No comment.

Additional comments

Dear authors,

This is an interesting work to investigate the differentially expressed genes in gliomas. Nevertheless, the different types of gliomas, including low grade gliomas and high grade gliomas were not properly introduced. Furthermore, it is not clear how the discussion fits into the overall aim of the study apart from the title. It would seem that the focus on SAA1 and TIMP1 individual genes, without discussing the roles and correlation of the other hub genes with higher degree of connectivity, can be further enhanced in the paper.

It was also noticed at many instances in the paper, the authors did not carefully use the abbreviation for example in the cases of Glioblastoma multiforme, Glioblastoma, GBM, Low Grade Glioma and LGG.

Reviewer 2 ·

Basic reporting

The manuscript is well written with enough backround information. Manuecript by Xu et al. is based on bioinformatics analysis of differentially expressed genes between glioblastomas and low-grade gliomas. The manuscript is well defined with enough background, I only cannot find the legends to the figures. Hypothesis, figures and tables are well written.

Experimental design

I miss the use of more expression datasets from GEO database, namely authors took only two data sets, GSE52009 and GSE53733. With more datasets they could get more reliable results with higher number of samples.

To state that TIMP-1 and SAA1 could be used as biomarkers, the authors should validate those findings on cell lines or tumor tissues using qPCR/Western blot/IHC or flow cytometry or something similar.

Validity of the findings

In Figure 4, the whole p values are not listed (only 0) and then you cannot know if the result are statistically significat. Please provide p -value, is the p value less then 0.05, 0.01, 0.001?

Reviewer 3 ·

Basic reporting

This manuscript is a professional article. No further comments.

Experimental design

The authors applied bioinformatic tool to analyze the GBM and low grade glioma data from GEO database. Study identified SAA1 and TIMP signaling pathway as plausible biomarker to distinguish low grade and malignant glioma. Authors also propose the contribution of SAA1 and TIMP1 signaling in GBM development.

Validity of the findings

In general, the analysis was good and the descriptions are sufficient to support the results.However, there are some points need to be answer. First, the authors should verify their finding with other data base to see if their finding can be confirm in other database. Second, the role and finding of SAA1 could be very different in different GBM cell lines from previous publications. Such as in Knebel's study, they found dual role of SAA1 to GBM progression; could authors find any correlation or association to explain/distinguish this? Third, the databases are base on RNA expression level in GBM and LLG, which might not be able reflect to the peripheral level. Also, the different region of tumor tissues could expression different SAA1 and TIMP1. Authors could put this into discussion.

---

## Round 0.2 · accepted · Accept

Thank you for the submission of your revised manuscript which has undergone peer review and is now accepted for publication in PeerJ. Thank you.

# Reviewer 3 ·

Basic reporting

no comment

Experimental design

no comment

Validity of the findings

no comment

Additional comments

The authors have answer the questions from reviewers.